# Effect of Deferoxamine on Post-Transfusion Iron, Inflammation, and In Vitro Microbial Growth in a Canine Hemorrhagic Shock Model: A Randomized Controlled Blinded Pilot Study

**DOI:** 10.3390/vetsci10020121

**Published:** 2023-02-05

**Authors:** Melissa A. Claus, Lisa Smart, Anthea L. Raisis, Claire R. Sharp, Sam Abraham, Joel P. A. Gummer, Martin K. Mead, Damian L. Bradley, Rachel Van Swelm, Erwin T. G. Wiegerinck, Edward Litton

**Affiliations:** 1School of Veterinary Medicine, Murdoch University, Murdoch, WA 6150, Australia; 2Perth Veterinary Specialists, Osborne Park, WA 6017, Australia; 3Small Animal Specialist Hospital, Tuggerah, NSW 2259, Australia; 4Forensic Sciences Laboratory, ChemCentre, Resources and Chemistry Precinct, Bentley, WA 6102, Australia; 5School of Science, Edith Cowan University, Joondalup, WA 6027, Australia; 6Intensive Care Unit, Rockingham General Hospital, Cooloongup, WA 6168, Australia; 7Hepcidinanalysis.com, Department of Laboratory Medicine, Translational Metabolic Laboratory (TML 830), Radboud University Medical Center, 6525 Nijmegen, The Netherlands; 8Intensive Care Unit, Fiona Stanley Hospital, Murdoch, WA 6150, Australia; 9School of Medicine, University of Western Australia, Crawley, WA 6009, Australia

**Keywords:** hepcidin, non-transferrin-bound iron, labile plasma iron, desferrioxamine, storage lesion, chelation

## Abstract

**Simple Summary:**

Blood transfusions can be lifesaving but can also harm patients by causing inflammation and increasing the risk of infection. Harm may occur from increasing the amount of unbound iron in circulation. When it is not bound to special carriers, iron is toxic, causing inflammation and supporting bacterial growth. This study aimed to determine if giving deferoxamine, a drug designed to bind iron, just after a transfusion would prevent the increase in unbound iron and inflammation that occurs in dogs following blood transfusions. All dogs in the study had increased unbound iron levels and markers of inflammation after receiving a transfusion. However, when unbound iron levels and markers of inflammation were compared between dogs that received deferoxamine and dogs that received a placebo, there were no differences detected. Furthermore, deferoxamine did not slow the growth of bacteria within blood samples taken from dogs during the study compared to placebo. In conclusion, the dose of deferoxamine used in this study did not prevent inflammation in the transfused dogs nor did it inhibit bacterial growth in blood samples from these dogs.

**Abstract:**

Red blood cell (RBC) transfusion is associated with recipient inflammation and infection, which may be triggered by excessive circulating iron. Iron chelation following transfusion may reduce these risks. The aim of this study was to evaluate the effect of deferoxamine on circulating iron and inflammation biomarkers over time and in vitro growth of *Escherichia coli* (*E. coli*) following RBC transfusion in dogs with atraumatic hemorrhage. Anesthetized dogs were subject to atraumatic hemorrhage and transfusion of RBCs, then randomized to receive either deferoxamine or saline placebo of equivalent volume (*n* = 10 per group) in a blinded fashion. Blood was sampled before hemorrhage and then 2, 4, and 6 h later. Following hemorrhage and RBC transfusion, free iron increased in all dogs over time (both *p* < 0.001). Inflammation biomarkers interleukin-6 (IL6), CXC motif chemokine-8 (CXCL8), interleukin-10 (IL10), and keratinocyte-derived chemokine (KC) increased in all dogs over time (all *p* < 0.001). Logarithmic growth of *E. coli* clones within blood collected 6 h post-transfusion was not different between groups. Only total iron-binding capacity was different between groups over time, being significantly increased in the deferoxamine group at 2 and 4 h post-transfusion (both *p* < 0.001). In summary, while free iron and inflammation biomarkers increased post-RBC transfusion, deferoxamine administration did not impact circulating free iron, inflammation biomarkers, or in vitro growth of *E. coli* when compared with placebo.

## 1. Introduction

Transfusion of red blood cells (RBCs) can be lifesaving; however, RBC transfusion is independently associated with increased morbidity and mortality in people [1,2,3,4]. These findings may be due, in part, to storage lesions that result in progressive detrimental changes to the integrity of the stored RBCs over time [5,6,7,8,9,10,11]. Breakdown of ageing RBCs increases iron-rich, cell-free hemoglobin in RBC units and, subsequently, unbound iron concentration in transfusion recipients [12,13,14,15].

Rapid increases in serum iron can overwhelm transferrin-binding capacity, resulting in reactive free-iron species. Non-transferrin-bound iron (NTBI)—and, particularly, labile plasma iron (LPI)—generates free radicals [16,17]. These damage cellular structures and activate cytokine transcription factors, both of which induce inflammation and can lead to tissue and organ dysfunction [16,18,19]. Free iron may also increase risk of infection by encouraging the growth of pathogenic bacteria, such as *Escherichia coli* (*E. coli*) [20]. Physiological mechanisms are activated to decrease iron availability during sickness, which may decrease susceptibility to pathogens. However, these mechanisms may not be useful in acute iron overload [21]. Interventions aimed at binding NTBI and LPI associated with RBC transfusion may, therefore, mitigate transfusion-induced oxidative stress, systemic inflammation, and infection risk.

Deferoxamine is a chelating agent approved for clinical use in people and used off-label in veterinary medicine. Deferoxamine binds free iron and removes iron from transferrin and ferritin, freeing these proteins to bind additional free iron [22]. While deferoxamine is an established treatment in people and companion animals for acute ingestion-related iron poisoning and in people for chronic iron overload associated with transfusion dependence, there is limited data on its ability to decrease serum iron levels associated with single RBC transfusions [23]. In a murine study of iron chelation, deferoxamine inhibited extra-intestinal pathogenic *E. coli* growth when added to pooled plasma from mice that had been transfused with stored blood [24]. Whether IV deferoxamine therapy after RBC transfusion reduces serum iron levels in vivo, and whether this is associated with a decrease in inflammatory markers or the ability of the recipients’ blood to support in vitro bacterial growth remains uncertain.

The objective of this study was to evaluate the effect of deferoxamine on circulating iron and inflammation biomarker concentrations over time and on in vitro *E. coli* growth following RBC transfusion in dogs with atraumatic hemorrhage. The primary hypothesis was that NTBI would be lower in dogs treated with deferoxamine than in dogs treated with placebo.

## 2. Materials and Methods

### 2.1. Dogs

The study protocol was approved by the university’s institutional animal ethics committee (R2732/15). This study included 20 healthy, transfusion-naive ex-racing greyhounds donated to the Murdoch University School of Veterinary Medicine for euthanasia. It was performed over the course of a 13 month period based on the availability of the donated greyhounds and canine RBC bags in the veterinary blood bank. The median age of the dogs was 2.5 years (range: 1.5–6 years), the median weight was 28.75 kg (range: 24–35 kg), and 9 of the 20 dogs were male. The dogs were determined to be healthy based on a physical examination performed by an experienced staff veterinarian (MAC or CRS). The physical examination included visual inspection of all parts of the dog, auscultation of the heart and lungs, and palpation of the abdomen, peripheral lymph nodes, and limbs for any abnormalities.

### 2.2. Study Protocol

Each dog was sedated with intramuscular methadone (0.3 mg/kg body weight (bw), Ilium methadone, Troy Laboratories Pty Ltd., Glendenning, NSW, Australia). Following intravenous cannula (20G × 1.88-inch BD Insyte IV cannula, Becton Dickinson Pty Ltd., Macquarie Park, NSW, Australia) placement in a cephalic vein, anesthesia was induced with intravenous alfaxalone (1–3 mg/kg bw, Alfaxan Multidose, Jurox Animal Health, Rutherford, NSW, Australia) to enable orotracheal intubation (10 mm internal diameter endotracheal tube, Global Vet Products, Chermside West, QLD, Australia). General anesthesia was maintained with delivery of isoflurane (inspired concentration 1.5% to 2%, Isothesia NXT, Piramal Enterprises Ltd., Telangana, India) in oxygen using a rebreathing anesthetic circuit (GE Datex Ohmeda Aestiva 5, GE Healthcare, Wauwatosa, WI, USA). Following anesthesia, an additional intravenous cannula (18G × 1.88-inch BD Insyte IV cannula, Becton Dickinson Pty Ltd., Macquarie Park, NSW, Australia) was placed in the contralateral cephalic vein.

The dogs were placed in left lateral recumbency, and a surgical approach to the left femoral or right carotid artery was performed. A 16 gauge over-the-needle cannula (Angiocath, Becton Dickinson Pty Ltd., Macquarie Park, NSW, Australia) was inserted into the artery and secured. The cannula was connected to a three-way tap and a pressure transducer for invasive arterial blood pressure monitoring (Meritrans DTXPlus, Merit Medical, Singapore Pte Ltd., Singapore). Pulse oximetry, capnography, electrocardiography, and direct arterial pressure were monitored (SurgiVet Advisor 3 Parameter Vital Signs Monitor, Smiths Medical, London, UK) throughout the experiment. All dogs were mechanically ventilated with a volume-controlled, time-cycled ventilator mode to maintain end-tidal carbon dioxide between 35 and 40 mmHg. Compound sodium lactate (Compound sodium lactate (Hartmann’s solution), Baxter Healthcare Pty Ltd., Toongabble, NSW, Australia) and fentanyl (Fentanyl GH, Generic Health Pty Ltd., Box Hill, VIC, Australia) were intravenously infused at 10 mL/kg bw/hr and 2 µg/kg bw/hr, respectively, throughout the experiment.

Randomization was performed prior to the start of the study using a computer-generated, randomly derived sequence. Allocation concealment was maintained using sequentially numbered, sealed, opaque envelopes containing a numeric code for the study arm to which the dog was randomized. At the time of enrollment, the body weight of the dog and the next sequential sealed envelope containing the group allocation were given to an unblinded veterinarian external to the study to prepare the study infusion for administration. The deferoxamine group had the study infusion prepared in such a way as to deliver a total dose of 13.3 mg/kg bw, which was added to 0.9% *w*/*v* NaCl (Sodium chloride 0.9% IV Infusion, Baxter Healthcare Pty Ltd., Toongabble, NSW, Australia) to bring the total volume for infusion to 20 mL. This dose of deferoxamine (Desferal, Novartis, Macquarie Park, NSW, Australia) was determined according to the expected iron load administered to dogs during transfusion and the iron-binding capacity of deferoxamine [25,26,27,28]. The control group had the study infusion prepared in such a way as to deliver a total volume of 20 mL of 0.9% *w*/*v* NaCl alone. To ensure blinding, all syringes and giving sets were sheathed in an opaque cover prior to handing them over to the researchers.

Controlled, atraumatic hemorrhage was achieved by the aseptic collection of two 450 mL units of blood removed sequentially over a total of 15 min from the femoral or carotid artery cannula into blood collection bags (Fenwal Whole Blood CPDA-1 Triple Blood-Pack Unit, Fenwal Inc, Lake Zurich, IL, USA) for use in the veterinary hospital blood bank. Immediately following controlled hemorrhage, two allogenic, DEA 1-negative or type-matched non-leukoreduced packed RBC units with the additive saline, adenine, glucose, and mannitol, sourced from the veterinary hospital blood bank and aged between 35 and 43 days, were administered simultaneously through both cephalic cannulas using infusion pumps to deliver a total dose of 15 mL/kg bw over a period of 20 min. Immediately following transfusion, the previously prepared study infusion was administered using a syringe pump through one of the cephalic cannulas over a period of 55 min. Other than blood sample collection, no additional interventions occurred until the dogs were euthanized with a total volume of 20 mL of sodium pentobarbital (Lethabarb, Virbac Australia Pty Ltd., Milperra, NSW, Australia) through a cephalic cannula at the 6 h post-baseline time point (Figure 1).

### 2.3. Sample Collection and Analysis

Baseline blood samples were collected just prior to controlled hemorrhage and then at two, four, and six hours after baseline. Blood samples were collected into either a clot activator (Serum 4 mL Vacutainer tubes, Becton Dickinson Pty Ltd., Macquarie Park, NSW, Australia) or lithium heparin tubes (Lithium heparin 4 mL Vacutainer tubes, Becton Dickinson Pty Ltd., Macquarie Park, NSW, Australia) at each time point and centrifuged for 10 min at 3000× *g*, and serum and plasma were stored at −80 °C for later batched analysis. Blood was also collected into EDTA tubes (EDTA 4 mL Vacutainer tubes, Becton Dickinson Pty Ltd., Macquarie Park, NSW, Australia) for full blood counts at each time point (XT-2000i analyser, Sysmex, Kobe, Japan, at Vetpath Laboratory Services, Ascot, WA, Australia). Finally, an additional 100 mL of blood was aseptically collected into serum separator tubes (SST 8.5 mL Vacutainer tubes, Becton Dickinson Pty Ltd., Macquarie Park, NSW, Australia) at the 6 h time point and centrifuged for 10 min at 3000× *g*. The serum was aseptically decanted into sterile conical tubes (Nunc, Thermo Fischer Scientific, Waltham, MA, USA) and used for microbial growth studies (see Section 2.4).

Total serum iron, canine-specific ferritin, and total iron-binding capacity (TIBC) were assessed at a commercial laboratory (Kansas State Veterinary Diagnostic Laboratory, Manhattan, KS, USA). Ferritin was measured with an enzyme-linked immunosorbent assay using mouse monoclonal anti-canine ferritin IgG antibody with a sandwich technique. Spectophotometry was used to quantify total serum iron and unbound iron-binding capacity. The calculated sum of these two measurements represented the total iron-binding capacity.

Non-transferrin-bound iron and LPI were measured at a commercial laboratory (Radboudumc Center for Iron Disorders, Nijmegen, Netherlands). Prior to examining the study samples, both assays were first adapted for use with canine serum. Mobilization of serum NTBI was achieved by chelating it with nitrilotriacetate and then separating it from transferrin-bound iron by ultrafiltration and measuring it spectrophotometrically at 540 nm after the addition of the iron-reducing agent thioglycolic acid and bapthophenanthroline disulfonic acid, which forms a red complex with ferrous ion [29]. Labile plasma iron analysis was based on the measurement of the redox-active and readily chelatable fraction of NTBI. Specifically, the assay measured the generation of iron-catalyzed radicals in the presence of a low ascorbate concentration. Formation of radicals was measured with the fluorgenic redox-sensitive probe dihydrorhodamine 123 [30].

Hepcidin was measured by mass spectrometry (MS) with a determined specificity for hepcidin-25 of the Canidae, as previously described [31]. Briefly, the hepcidin-25 peptide was isolated from serum and measured by liquid chromatography–mass spectrometry using an accurate mass quadrupole time-of-flight mass spectrometer (5600 TripleTOF q-TOF-MS, Sciex, Framingham, MA, USA). Quantitation was achieved against a synthetic hepcidin peptide standard (Product PLP-3785-PI, Peptides International, Louisville, KY, USA) as previously described [31].

Plasma interleukin-6 (IL6), CXC motif chemokine-8 (CXCL8), interleukin-10 (IL10), and keratinocyte-derived chemokine (KC) were measured in duplicate using a commercial magnetic bead multiplexed assay kit (Milliplex^®^ MAP Canine Cytokine Magnetic Bead Panel Kit, MilliporeSigma, Burlington, MA, USA) with a multiplexed biomarker analyzer (MAGPIX xMAP^®^, Luminex^®^ Corp, Austin, TX, USA) for samples from each time point. The sensitivity for each analyte was as follows: IL6: 3.7 pg/mL, CXCL8: 21.7 pg/mL, IL10: 8.5 pg/mL, and KC: 5.3 pg/mL. The standard curve range for all analytes was 12.2–50,000 pg/mL. Assays were performed according to the manufacturer’s instructions, which included censoring concentrations that were below the standard curve at the minimum detectable concentration.

### 2.4. In Vitro E. coli Growth

*Escherichia coli*, an extra-intestinal pathogen frequently isolated from septic patients, was chosen as a suitable bacterium for this study due to its requirement for iron to thrive [32,33]. *Escherichia coli* ATCC 25922 were cultivated on sheep blood agar (Thermo Fischer Scientific, Waltham, MA, USA) at 37 °C for 12–24 h prior to the start of the in vitro study. Colonies were inoculated into saline to achieve 0.5 McFarland units or an *E. coli* concentration of ~1.5 × 108 colony forming units (CFUs)/mL. This solution was diluted and inoculated into a fixed volume of canine serum (from the 6 h time point) in a sterilized glass Erlenmeyer flask to generate a final starting concentration of ~1.2 × 103 CFU/mL serum. The flask was capped with sterile cotton wool and aluminum foil and incubated for 24 h at 37 °C.

Samples were collected from the flask immediately and at 1, 2, 4, 6, 8, and 24 h after inoculation. Serial dilutions were performed at each time point, and 10 µL drops were plated on blood agar plates in triplicate. Plates were incubated at 37 °C for 8–12 h and *E. coli* growth was assessed by counting and averaging bacterial counts within each drop point at each dilution.

### 2.5. Statistical Analysis

Sample size estimation was calculated based on an assumption of a baseline NTBI of 2.95 μmol/L, an SD of 0.6, and an alpha of 0.05 [34], resulting in an estimate that ten dogs in each arm would provide 90% power to detect a conservative 30% decrease in NTBI in the deferoxamine arm (treatment group) compared with placebo (control group) [35].

Normality was assessed through visualization of histograms and Q–Q plots. Data are summarized as the mean (95% confidence interval) for normal data, geometric mean (95% confidence interval) for log-transformed data that approximated a normal distribution, and median (Q1–Q3) for non-normal data. Baseline characteristics were compared between groups using either Student’s *t*-test or Fisher’s exact test. Linear mixed-effects models were used to compare differences in outcome variables between groups, with time and treatment as fixed effects and dog as a random effect. Post hoc pairwise comparisons with *p* value correction for multiple comparisons were planned if there was significant interaction between time and treatment (*p* < 0.05). Given the lack of data in the literature on the kinetics of most of the measured variables in dogs, the effect of time alone (treatment and placebo group) within the model was also reported. For highly skewed data that could not be transformed to approximate a normal distribution, pairwise comparisons using Wilcoxon rank-sum tests between groups at each time point were performed. Data were analyzed using a commercial statistical software package (SAS 9.4, SAS Institute, Cary, NC, USA).

## 3. Results

### 3.1. Pre-Transfusion Characteristics

There were no differences in pre-transfusion characteristics between the treatment (*n* = 10) and control (*n* = 10) groups, including age (mean: 2.9 years (2.3–3.6) versus 3.1 years (2.1–4.1), respectively; *p* = 0.77), number of male dogs per group (*n* = 5 versus *n* = 4; *p* = 1.0), body weight (mean: 29.2 kg (26.3–32.0) versus 29.7 kg (27.8–31.6); *p* = 0.73), and volume of blood removed during the hemorrhage phase (mean: 30.9 mL/kg (27.9–33.9) versus 29.9 mL/kg (28.0–31.8); *p* = 0.53). There was no significant difference in the change in hematocrits over time (*p* = 0.95) between groups (Table 1).

### 3.2. Iron Parameters

Total iron-binding capacity was significantly higher at T2 and T4 (both *p* < 0.001; Wilcoxon rank-sum) in the treatment group compared to the control group (Figure 2A), reflecting circulating deferoxamine. Otherwise, there were no significant differences between groups in the change over time for the other measures of iron, including NTBI (*p* = 0.38), total serum iron (*p* = 0.39), LPI (*p* = 0.09), serum ferritin (*p* = 0.60), and hepcidin (*p* = 0.89) concentrations (Figure 2B–F). When assessing changes over time across all dogs, NTBI, total serum iron, LPI, serum ferritin, and hepcidin concentrations all increased significantly over time (all *p* < 0.001).

### 3.3. Inflammation Biomarkers

There were no significant differences between groups in the changes in inflammation biomarker concentrations for IL6 (*p* = 0.91), CXCL8 (*p* = 0.92), IL10 (*p* = 0.41), and KC (*p* = 0.43) over time (Figure 3). When assessing change over time across all dogs, IL6, IL10, and KC concentrations increased significantly over time (all *p* < 0.001), whereas CXCL8 concentrations decreased significantly over time (*p* < 0.001). There were no significant differences in the changes over time in neutrophil (*p* = 0.18), lymphocyte (*p* = 0.89), or monocyte (*p* = 0.10) counts between groups. Across all dogs, there was a significant increase in the neutrophil count over time (*p* = 0.041) and a decrease in the lymphocyte count over time (*p* < 0.001) (Table 1).

### 3.4. In Vitro E. coli Growth

Assessment of *E. coli* growth in serum in the treatment and control groups demonstrated no significant difference in bacterial growth over time between groups (*p* = 0.27). Across groups, *E. coli* growth increased significantly over time (*p* < 0.001), with typical lag and log phases of growth between 2 and 8 h (Figure 4).

## 4. Discussion

In this blinded, randomized, placebo-controlled study assessing the effect of deferoxamine on iron levels, inflammation, and in vitro microbial growth in dogs undergoing transfusion of stored RBCs after atraumatic hemorrhage, NTBI, LPI, hepcidin, and inflammation biomarkers increased over time post-transfusion in all dogs. However, there was no demonstrable effect of deferoxamine administration on free iron, inflammation biomarkers, or in vitro microbial growth.

In this study, there was an early rise in free iron levels and a later rise in inflammation biomarker concentrations in dogs after hemorrhage and RBC transfusion. While it is understood that chronic iron overload impairs organ function—in particular, the liver, heart, and endocrine glands [36,37,38]—there are fewer studies assessing the pathophysiological sequelae of acute increases in free iron. Redox-active LPI and NTBI have been shown in cell culture models to generate free radicals both intra- and extracellularly [16,17,39,40]. Additionally, there is evidence that excess free iron generates reactive oxygen species, which contribute to the promotion of inflammation, lipid peroxidation, and tissue damage [19,38,41,42,43]. Notably, these findings are consistent with the observed effects of augmented inflammation following older RBC transfusion in an experimental septic dog model [34]. Further studies are necessary to better explore the link between abrupt increases in free iron and exacerbation of inflammation in critically ill individuals.

Hepcidin also significantly increased over time following transfusion in this study. Hepcidin is a hepatocellular peptide hormone that promotes internalization and degradation of ferroportin channels in a variety of cell types to decrease circulating iron [31,44,45,46]. There are many physiological mechanisms that contribute to changes in hepcidin production. First, erythropoiesis acts as a potent suppressor of hepcidin production via a recently discovered hormone, erythroferrone. In the presence of erythropoietin, erythroblasts secrete erythroferrone, which acts on hepatocytes to suppress production of hepcidin [47]. Conversely, when erythropoiesis is suppressed and erythroferrone levels are low, as occurs following a blood transfusion and during critical illness, hepcidin concentrations increase. Under these conditions, hepcidin concentration may predict response to iron therapy and may determine which patients will benefit from erythropoiesis-stimulating therapy [48,49]. A study in healthy volunteers found that serum hepcidin concentrations rise quickly after transfusion, peaking at around 12 h [50]. Transfusion-associated increases in hepcidin have also been demonstrated in pre-term neonates [51,52,53] and patients with gastric cancer [54] and β-thalassemia major [55]. Second, increases in circulating iron increase hepcidin production. As plasma iron levels increase, a greater percentage of transferrin becomes saturated. Iron-saturated transferrin binds to transferrin receptors on hepatocytes, which appears to rapidly increase hepcidin production [56,57]. Third, inflammation—and, specifically, the inflammatory cytokine IL6—stimulates hepatocytes to produce hepcidin [58]. The dogs in this study had all three stimulators of hepcidin production: RBC transfusion, elevated iron levels, and increased IL6. Thus, it is unsurprising that hepcidin increased over time. It is likely, however, that the peak hepcidin concentration in the dogs was missed due to completion of the study at 6 h.

The only difference between groups found in this study was a significant increase in TIBC at 2 and 4 h in the treatment group, reflecting the binding capacity of circulating deferoxamine. Clinically, this translates to iron chelation therapy preventing TIBC from accurately reflecting circulating transferrin levels. This finding is consistent with other studies in which deferoxamine increased TIBC and unbound iron-binding capacities [59,60]. Despite this evidence that sufficient deferoxamine was present within the blood to increase TIBC, there was no identifiable effect of deferoxamine on other outcomes in this study, such as free iron levels, markers of inflammation, or bacterial growth. There are two possible explanations for these findings: an insufficient dose of the study drug or insufficient duration of administration.

First, the dose of deferoxamine in this canine model was based on limited available data and may have been insufficient. The chosen dose was calculated based on assumptions that 1 mL of packed RBCs contains 0.8 mg of iron [25] and 10% of the volume of RBCs administered would immediately undergo hemolysis and release stored iron, based on results from a study of dogs that underwent transfusion of autologous 28 day old RBCs [26]. The blood administered, however, was 1 to 2 weeks older than the blood used in that study. Thus, it is possible that the advanced age of the RBCs may have led to a greater degree of hemolysis and release of iron post-transfusion. Additionally, the dogs in this study received uncrossmatched allogenic RBCs rather than autologous RBCs. While there were no clinical manifestations of acute hemolytic transfusion reactions in the dogs, incompatibility-induced hemolysis may have contributed to a greater liberation of iron post-transfusion than predicted. Indeed, many of the dogs had grossly hemolyzed plasma and serum at each blood collection time point post-transfusion.

Second, the duration of infusion of deferoxamine may have been insufficient to bind free iron released during the ongoing hemolysis of transfused RBCs. While the pharmacokinetics of deferoxamine in dogs is unknown, it is known to have a short half-life in people. In pharmacokinetic studies in humans, following a rapid IV bolus, unbound deferoxamine is only present for approximately 20 min, after which it has bound free iron and becomes ferrioxamine. It is then fully eliminated from the body within 3 h [37]. In a study assessing the kinetics of deferoxamine and NTBI in β-thalassemia patients, NTBI was found to rapidly increase in circulation within 30 min of discontinuing a continuous intravenous infusion of deferoxamine [35]. That time frame is the same length of time between when the deferoxamine was discontinued and when the first blood sampling time point post-baseline occurred in this study. Thus, if the canine pharmacokinetics of deferoxamine are similar to humans, any decrease in NTBI or LPI that would have occurred with deferoxamine administration post-transfusion may have been missed. Future studies are needed to determine if higher doses, repeated doses, or longer infusion times for deferoxamine would have significant effects on post-transfusion free iron levels or inflammation.

This study has some limitations. While all dogs appeared healthy on examination, occult inflammation or other disease may have led to the variability in measured iron levels and inflammation biomarkers among the dogs. This variability in the small sample size may have contributed to a type II error. This model was quite short in duration, precluding us from finding any potential difference between groups that may have occurred at a later time point. The paucity of information in the literature regarding deferoxamine dosing likely led to the use of an inappropriately low dose for too short a time, precluding us from identifying a difference between groups. This model was employed with healthy subjects that received a conservative dose of RBCs, whereas an effect of deferoxamine on post-transfusion free iron and inflammation may be more apparent in critically ill subjects, especially those receiving massive transfusion. Nonetheless, the increase in free iron and inflammatory markers across all dogs in this study validates the model for future investigations. Finally, the RBCs used were non-leukoreduced, which can potentiate the storage lesion and increase inflammatory mediators within the product. This could confound the ability to differentiate inflammation from excess free iron from that caused by other inflammatory mediators within the stored RBC unit.

## 5. Conclusions

In conclusion, NTBI, LPI, hepcidin, IL6, IL10, and KC increased following controlled hemorrhage and transfusion of RBC. Deferoxamine, when administered as a short infusion, did not significantly impact circulating free iron levels, inflammation biomarkers, or the ability of recipient blood to support in vitro *E. coli* growth when compared with placebo. Further studies are required to determine if higher deferoxamine doses or longer infusion times would mitigate increases in free iron levels and resulting inflammation.

Overall, while the results from this study failed to show a positive impact of deferoxamine on transfusion recipients, publication of the experimental design and achieved results from this pilot study is important to inform future research on post-transfusion inflammation and specific treatments that may mitigate this inflammatory response.

## Figures and Tables

**Figure 1 vetsci-10-00121-f001:**
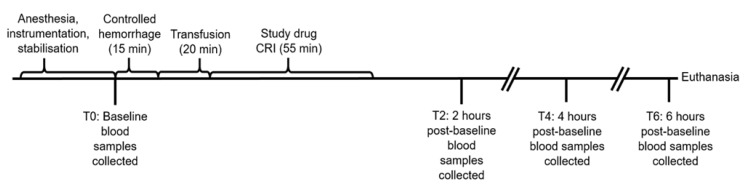
Timeline of experimental model. Baseline blood samples included EDTA, lithium heparin, and serum at all time points and an additional quantity of serum for microbial studies at T6. Controlled hemorrhage involved the removal of 900 mL of blood from the arterial catheter. Transfusion involved 15 mL/kg of allogenic, stored, packed red blood cells. The study drugs were deferoxamine in 20 mL of saline (treatment group, *n* = 10) or 20 mL of saline alone (control group, *n* = 10).

**Figure 2 vetsci-10-00121-f002:**
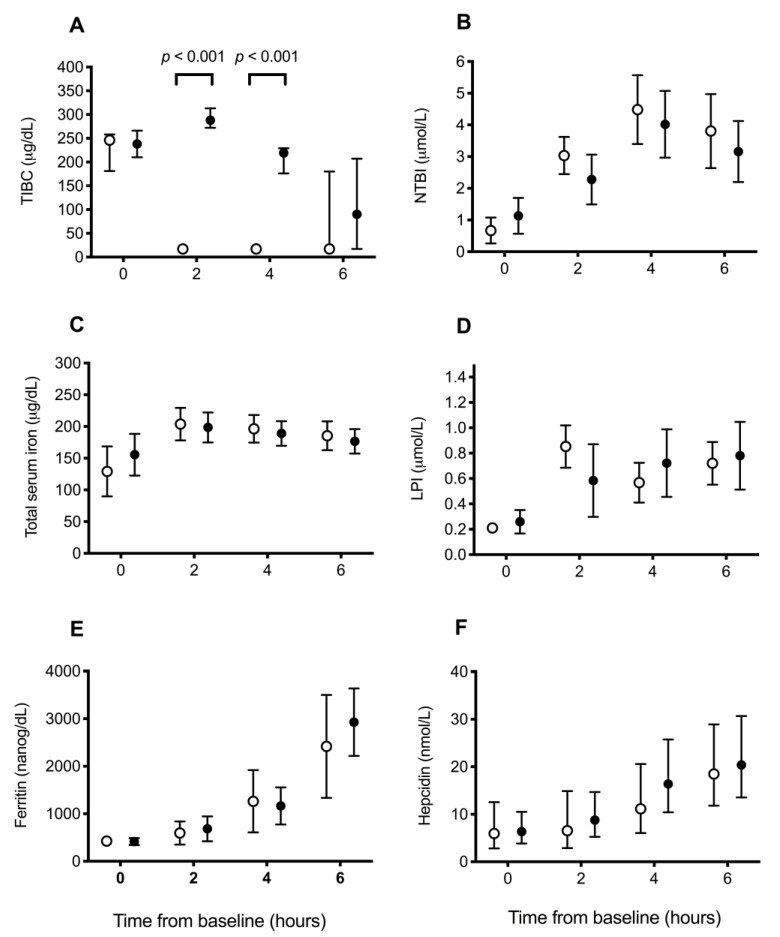
Blood iron parameters in dogs receiving saline or deferoxamine at baseline and 2 hourly intervals after atraumatic hemorrhage and allogenic red blood cell transfusion. Mean (95% CI) of (**A**)) total iron-binding capacity (TIBC), (**B**) non-transferrin-bound iron (NTBI), (**C**) total serum iron, (**D**) labile plasma iron (LPI), (**E**) ferritin, and (**F**) hepcidin in dogs receiving saline (control: ○, *n* = 10) or deferoxamine (treatment: ●, *n* = 10). Between baseline and 2 h, all dogs had 900 mL of whole blood removed, had 15 mL/kg of allogenic, stored, packed red blood cells transfused over 20 min, and then received 20 mL of saline (control) or an equal volume of 13.3 mg/kg deferoxamine in saline (treatment) as a continuous intravenous infusion lasting 55 min.

**Figure 3 vetsci-10-00121-f003:**
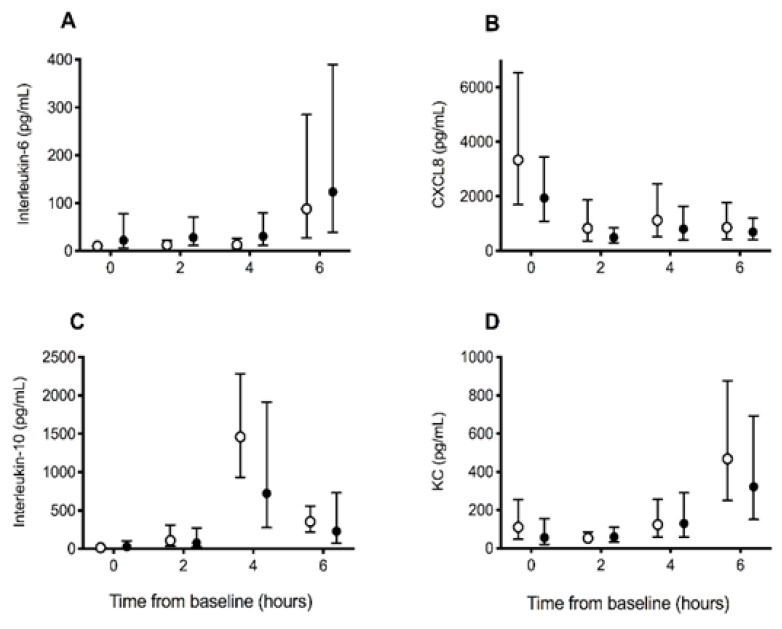
Inflammation biomarker concentrations in dogs receiving saline (control) or deferoxamine (treatment) at baseline and 2 hourly intervals after atraumatic hemorrhage and allogenic red blood cell transfusion. Means (95% CI) of (**A**) interleukin-6, (**B**) CXC motif chemokine-8 (CXCL8), (**C**) interleukin-10, and (**D**) keratinocyte-derived chemokine (KC) in dogs receiving saline (control: ○, *n* = 10) or deferoxamine (treatment: ●, *n* = 10). Between baseline and 2 h, all dogs had 900 mL of whole blood removed, had 15 mL/kg of allogenic, stored, packed red blood cells transfused over 20 min, and then received 20 mL of saline (control) or an equal volume of 13.3 mg/kg deferoxamine in saline (treatment) as a continuous intravenous infusion lasting 55 min.

**Figure 4 vetsci-10-00121-f004:**
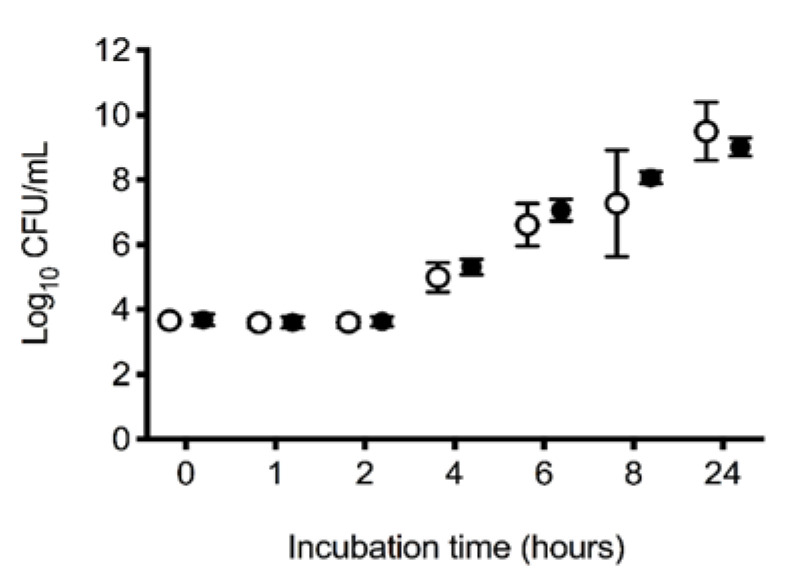
In vitro 24 h log_10_ growth of *Escherichia coli* ATCC 25922 in serum. Mean (95% CI) in vitro 24 h log_10_ growth (colony forming units; CFUs) of *Escherichia coli* ATCC 25922 in serum removed from dogs at the completion of a 6 h experiment. Between baseline and the second hour of the 6 h experiment, all dogs had 900 mL of whole blood removed, had 15 mL/kg of allogenic, stored, packed red blood cells transfused over 20 min, and then received 20 mL of saline (control: ○, *n* = 10) or an equal volume of 13.3 mg/kg deferoxamine in saline (treatment: ●, *n* = 10) as a continuous intravenous infusion lasting 55 min.

**Table 1 vetsci-10-00121-t001:** Serial hematocrits and white blood cell counts in dogs receiving saline (control, *n* = 10) or deferoxamine (treatment, *n* = 10) at baseline and 2 hourly intervals after atraumatic hemorrhage and allogenic red blood cell transfusion. Data presented are means (95% confidence interval).

	Baseline	2 Hours ^1^	4 h	6 h	*p* Value *
Hematocrits (L/L)					
Control	0.44 (0.41–0.47)	0.44 (0.42–0.46)	0.41 (0.39–0.43)	0.40 (0.38–0.42)	0.95
Treatment	0.46 (0.44–0.48)	0.47 (0.45–0.50)	0.43 (0.41–0.45)	0.42 (0.40–0.45)	
White blood cells (×10^9^/L)					
Control	4.73 (3.54–5.92)	4.97 (3.68–6.26)	3.96 (2.95–4.97)	4.58 (2.67–6.49)	0.23
Treatment	4.03 (3.35–4.71)	4.52 (3.50–5.54)	4.83 (3.98–5.68)	5.50 (4.50–6.50)	
Neutrophils (×10^9^/L)					
Control	3.43 (2.46–4.40)	3.73 (2.71–4.76)	2.98 (2.14–3.82)	3.89 (2.10–5.70)	0.18
Treatment	2.81 (2.21–3.40)	3.48 (2.56–4.41)	3.91 (3.16–4.66)	4.86 (3.90–5.83)	
Lymphocytes (×10^9^/L)					
Control	0.99 (0.68–1.29)	0.90 (0.71–1.09)	0.70 (0.55–0.85)	0.50 (0.36–0.63)	0.89
Treatment	0.99 (0.87–1.10)	0.79 (0.62–0.95)	0.62 (0.51–0.73)	0.46 (0.35–0.57)	
Monocytes (×10^9^/L)					
Control	0.17 (0.11–0.23)	0.22 (0.10–0.34)	0.18 (0.12–0.24)	0.13 (0.06–0.21)	0.10
Treatment	0.12 (0.10–0.14)	0.14 (0.11–0.17)	0.21 (0.18–0.25)	0.14 (0.09–0.19)	

^1^ Between baseline and 2 h, all dogs had 900 mL of whole blood removed, had 15 mL/kg of stored, packed red blood cells transfused over 20 min, and then received 20 mL of saline (control) or an equal volume of 13.3 mg/kg deferoxamine in saline (treatment) as a continuous intravenous infusion lasting 55 min. * *p* value represents the difference between groups over time.

## Data Availability

Data supporting the reported results can be requested by emailing the corresponding author, Melissa Claus, at dr.claus@gmail.com.

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
