# Peer review of "Effect of Deferoxamine on Post-Transfusion Iron, Inflammation, and In Vitro Microbial Growth in a Canine Hemorrhagic Shock Model: A Randomized Controlled Blinded Pilot Study"

_vetsci, 2023, doi:10.3390/vetsci10020121_

Round 1

Reviewer 1 Report

This paper focuses on iron poisoning during blood transfusion in dogs, and evaluates the effects of  a common iron chelator, deferoxamine,using retired racing dogsIron toxicity is a clinical problem in patients requiring repeated blood transfusion therapy and is often treated in medicine with other oral iron chelators. Acute iron poisoning in a single blood transfusion is rarely a clinical problem, but it is a potential hazard when degraded blood products are used. The authors focused on this point and tried to verify the necessity and usefulness of deferoxamine. However, the expression of iron toxicity in model animals has not been successful, and therefore the efficacy of the drug has not been verified. It is scientifically written in an easy-to-understand manner, but it is not possible to judge whether it will arouse the reader's interest.

The points to note are as follows.

Major:

-Please describe the validity of the measured inflammatory markers. In dogs, I think that CRP measurement is common clinically. What is the reason for measuring IL-6, 10, etc. without using CRP? The blood collection time for this experiment is up to 6 hours. I think the time course of this study to assess the change of inflammatory markers is too short. Please describe why you designed the experiment in this way.

Minor point:

-L194 "540nM" might be "540nm"

Reviewer 2 Report

In this very interesting manuscript, Claus et al presented a beautiful set of data that suggests that deferoxamine did not prevent inflammation in the transfused dogs nor did it inhibit bacterial growth in blood samples from these dogs compared to placebo controls. The study design is thoughtful and well-performed, however, I have a major concern about this study that this is a negative dataset and in that respect it is not moving the field forward. To get rid of this negativity, the authors need to select few other doses of the drug to get a treatment condition when it would show some effect (as they have currently mentioned in the conclusion section). There is no way this can be skipped as future work as they have tried to do. Also, why did the authors used deferoxamine in the first place? Can they combine it with some other iron-chelating drugs? I think this study would have a lot of impact on transfusion-related therapeutic approaches only if they can show some positive effects of the drug used in the study either alone or with some other combination of drugs. In my opinion, without these studies, the authors may need to boost the impact of the study in the conclusion section by mentioning why at this current stage this study is impactful enough and how it is progressing the field. currently, as an outside reader, it is not clear at all to appreciate the strength of the study. Unfortunately, without any of these, I would not recommend publishing this article in MDPI-Veterinary Sciences.

Reviewer 3 Report

Dear authors,

The present manuscript is well written and presented. 

My minor suggestions are related with the eligible criteria of the participants. It should be mentioned if there was a previous transfusion in the medical history of the animals. Moreover, it should be mentioned if the participants were tested for diseases that could affect the hematopoietic system (Ehrichia, dirofillaria etc)

Finally, it should be mentioned if there is any discussion/concern about the blood type of the donor and the receiver.

Round 2

Reviewer 2 Report

I thank the authors for taking time to respond. I agree that negative data should not be neglected with regard to publication, rather should be judged in terms of its impact to make "go" or "no go" decisions for future experiments. I thought, as such, although the manuscript contains beautiful piece of data, its impact looking forward to future studies was not reflected in the written form. I would like to see few of the justifications that the authors wrote in the rebuttal to be included in the conclusion section, which I am not sure if they have included in the current version. I would ask them to include exactly those rebuttal lines that they put in the response.

I believe an article published in a journal not only talks about itself, but also, carries a huge weight in terms of boosting the impact factors of the journal as a whole. In that regard, any study that would be published in future has to justify its impact for significantly upcoming journals like Veterinary Sciences. 

Author Response

Many thanks for your very constructive feedback. We are grateful for your comments and have added the final sentence of the conclusion as an attempt to meet your request. Please know there are more details in the limitations section of the discussion that speak specifically to the small sample size and the likely insufficient dose of deferoxamine administered to this population.

"In conclusion, NTBI, LPI, hepcidin, IL6, IL10 and KC increased following controlled hemorrhage and transfusion of RBC. Deferoxamine, when administered as a short infusion, did not significantly impact circulating free iron levels, inflammation biomarkers, or the ability of recipient blood to support in vitro E. coli growth when compared with placebo. Further studies are required to determine if higher deferoxamine doses or longer infusion times mitigates increases in free iron levels and resulting inflammation.

Overall, while the results from this study failed to show a positive impact of deferoxamine on the transfusion recipients, publication of the experimental design and achieved results from this pilot study is important to inform future research on post-transfusion inflammation and specific treatments that may mitigate this inflammatory response."